# Life Cycle Transitions in the Freshwater Jellyfish *Craspedacusta sowerbii*

**DOI:** 10.3390/biology13121069

**Published:** 2024-12-20

**Authors:** Kent Winata, Jonathan A. Zhu, Katherine M. Hanselman, Ethan Zerbe, Jensyn Langguth, Nadine Folino-Rorem, Paulyn Cartwright

**Affiliations:** 1Department of Ecology and Evolutionary Biology, University of Kansas, Lawrence, KS 66045, USA; kentwinata@ku.edu (K.W.); katherinehanselman@ku.edu (K.M.H.); 2Department of Biological and Health Sciences, Wheaton College, Wheaton, IL 60187, USA; jazhu@andrew.cmu.edu (J.A.Z.); nadine.rorem@wheaton.edu (N.F.-R.)

**Keywords:** cnidaria, hydrozoa, frustule, polyp, medusa, podocyst

## Abstract

*Craspedacusta sowerbii* is an invasive freshwater jellyfish that is found in all continents except Antarctica. *C. sowerbii* has a complex life cycle that includes several modes of asexual reproduction. While the medusa stage of the life cycle has been well documented, little is known of the environmental parameters conducive to the spread, growth and reproduction of the different life cycle stages. Here, we investigate laboratory conditions under which *C. sowerbii* can survive, reproduce and transition to different life cycle stages. This research provides insight into how an invasive species with a complex life cycle can potentially impact freshwater ecosystems.

## 1. Introduction

*Craspedacusta sowerbii* Lankester 1880 is a freshwater cnidarian belonging to the class Hydrozoa (Trachylinae, Limnomedusae, Olindiidae) [1]. Although cnidarians are primarily marine, invasions to fresh water have occurred multiple times, including the parasitic Myxozoa and *Polypodium* and the hydrozoans *Cordylophora, Hydra*, and freshwater jellyfish (including *C. sowerbii* and its close relatives) [2,3,4]. As in most hydrozoans, *C. sowerbii* has a complex life cycle alternating between a pelagic medusa (jellyfish) stage (Figure 1A and Figure 2E) and a benthic polyp stage (Figure 1C and Figure 2A) [5,6]. The polyp stage occurs on hard substrata such as rocks, wood and dreissenid mussels ([7], pers. observ.). The polyps are one to a few millimeters in size and reproduce asexually to form small colonies comprising 2–12 polyps that remain attached to each other at their base [6] (Figure 1D and Figure 2A). This is distinct from other hydrozoan colonies which tend to be much larger and are interconnected by tube-like structures called stolons [8]. Polyps bud medusae (jellyfish) from the polyp body column (Figure 1E and Figure 2D). These medusae swim in the water column and grow to a bell diameter of up to 1–3 cm and eventually become reproductive and spawn gametes [6] (Figure 1A and Figure 2E). Upon fertilization, the embryo develops into a planula larva (Figure 1B), which will eventually metamorphose into a polyp (Figure 1C), completing the life cycle [5,6].

*Craspedacusta sowerbii* has additional life cycle stages that are less typically found in a hydrozoan life cycle. This includes frustules, which are oval shaped bodies of tissue, resembling the planula larva, but bud from the polyps, crawl on the substrate and metamorphose into new polyps (Figure 1F and Figure 2B). A polyp or frustule can transform into a resting stage called a podocyst, which is a small bit of tissue covered by periderm (Figure 1G and Figure 2C). Podocysts have been reported to persist under extreme conditions, including desiccation and temperatures as low as 4 °C, and survive long periods of time [6,9].

The distribution of *Craspedacusta sowerbii* is widespread, thought to originate from the Upper Yangtze River basin in China, but inhabits every continent except Antarctica [10,11]. Although *C. sowerbii* has a wide distribution, the occurrence records are not as frequent in tropical regions as compared to northern and southern latitudes. Out of 3223 occurrences reported in the last 60 years, only 11 of those records were from the tropics [12]. *C. sowerbii* is thought to be spread by humans through the transport of aquatic plants and animals [13]. A recent study investigated the historical and predicted distributions of *C. sowerbii* [11]. Specifically, assembling occurrence records spanning 140 years from the literature and online databases, this study found that sightings have increased dramatically over the last 40 years, with North America and Europe having the most records at 1525 and 477, respectively. Using these occurrence records, Marchessaux et al. [11] utilized ecological niche modeling with inferred future climate conditions to predict distributions following climate change in the years 2050 and 2100. The model predicted that *C. sowerbii*’s current distribution will increase over 60% in all continents except Antarctica and Oceania. In addition, *C. sowerbii* is predicted to extend into northern latitudes in North America and Europe [11].

The documented distribution pattern is primarily based on the medusa stage. This is because *C. sowerbii* medusae are larger (approx. 3 cm at maturity) and more conspicuous relative to the other life cycle stages [6]. While it can be assumed that the other life cycle stages are present in the same environments as the medusae, very little is known about the distribution of the other life cycle stages. Because this species can asexually reproduce and disperse without medusae via frustules (Figure 1F and Figure 2B), and go into dormancy via podocysts (Figure 1G and Figure 2C), these other life cycle stages likely have a wider distribution than the medusa. This is supported with a study by Duggan and Eastwood [14], where they systematically sampled rocks for *C. sowerbii* polyps from lakes in New Zealand. They discovered polyps in 11 of 18 lakes where medusae had never been documented, suggesting that the actual distribution of *C. sowerbii* is more widespread than what has been documented from medusa observations [14].

For predicting effects of this invasive species on freshwater ecosystems, it is important to consider abundance as well as distribution. When observed in nature, *C. sowerbii* medusae tend to occur in ‘blooms’. Jellyfish blooms refer to the sudden appearance of a high density of individuals in a particular region [15]. Both freshwater and marine jellyfish blooms tend to be seasonal, with increases in numbers during times when water temperatures are warmer and food is most abundant. In *C. sowerbii*, medusae blooms tend to occur in the late summer to early fall, when water temperatures rise above 25 °C. Although medusae tend to only be found in warmer water, the blooms at any given location are usually infrequent and often unpredictable [16], suggesting that other ecological factors may be involved in *C. sowerbii* medusae production.

In marine jellyfish, longer term trends of increased numbers of both native and invasive populations have been documented [17]. Although the exact cause of these increases is unknown, it is speculated that climate change, eutrophication and overfishing are the primary drivers [17,18,19]. Increased abundance of jellyfish is of concern as they can become a nuisance to swimmers, reduce fish populations important for the fisheries industry, interfere with aquaculture and potentially clog power plants [18]. For *C. sowerbii* it has been shown that medusae occurrences are becoming more frequent as well [10]. This is particularly concerning given that the presence of medusae can have a dramatic effect on zooplankton community abundance [20,21,22].

Previous studies have shown that temperature and food availability play an important role in driving the success of *C. sowerbii* asexual and sexual reproduction [23,24,25,26]. Numerous experiments have been conducted to determine the effect of temperature on the growth of *C. sowerbii* polyps [3,25,27]. Although the optimal temperature seems to vary between studies, results suggest in general that *C. sowerbii* polyps grow and bud new polyps and frustules at temperatures between 19 °C and 29 °C. On the other hand, frustule production is suggested to be food-dependent, where increased food availability results in increased frustule production [24,25]. It has also been suggested that higher temperatures trigger medusa budding [6,22]. Successful rearing of medusae in the lab to the stage of gonad development have been reported; however, medusa failed to live to reproductive maturity [3,27,28].

This study aimed to contribute to this body of knowledge to better understand the environmental conditions in which each life cycle stage can grow and the cues that signal life cycle transitions. Previous studies have suggested genetic divergence between lineages of *Craspedacusta sowerbii* (e.g., [29]). Given the widespread distribution, and genetic divergences within this species complex, we first determined if our different lab strains belong to separate lineages and, if so, if these lineages responded differently to conditions for growth and life cycle stage transitions. Determining the optimal conditions for growth and life cycle transitions in different strains will help inform how we can better culture *C. sowerbii* for laboratory investigations. In addition, insight into the environmental triggers that promote life cycle transitions will enable us to better predict the potential negative effect *C. sowerbii* could impose on lake ecosystems.

## 2. Materials and Methods

### 2.1. Laboratory Strains

Laboratory strains from four different locations were used in this study: Panama (*C. sowerbii*_PAN); Japan (*C. sowerbii*_JPN); Lawrence, KS, USA (*C. sowerbii*_LKS) and Inverness, Illinois (*C. sowerbii*_INI). *C. sowerbii*_JPN was obtained from southern Haruke-ike (36.5167° N, 138.083° E) in Nagano Japan [30] *C. sowerbii*_LKS was collected from an artificial pond in 2022 in Lawrence, KS, USA (38.95° N, 95.2667° W). *C. sowerbii*_PAN was collected from the spillway of Gatun Lake (9.2641° N, 79.9314° W) from log pieces and small rocks in Gatun, Panama [3]. *C. sowerbii*_CCI and *C. sowerbii*_INI were collected by N. Folino-Rorem in 2022 from private ponds in Inverness, IL, USA (42.116540° N, 88.122060° W), by collecting rocks, twigs and pieces of wood.

### 2.2. Animal Care

*Craspedacusta sowerbii* polyps were maintained in hydra medium (HM) [31] either in 150 mm tissue culture dishes or glass finger bowls in aquaria with filtration at room temperature (22 °C). Polyps were fed two-day old *Artemia* nauplii three times a week to satiation. Media was changed after every feeding. Medusae were kept in small glass bowls (5.75″ diameter × 2.75″ high) in HM with gentle aeration. Medusae were hand fed two-day old *Artemia* nauplii daily for one week and then two-day old *Artemia* nauplii to satiation, by placing them in the glass dish. Photos from lab cultures of different stages in the life cycle are shown in Figure 2.

### 2.3. DNA Extraction and Sequencing

DNA was extracted from the frustule stage using the DNeasy Blood & Tissue Kit protocol (QIAGEN Hilden, Germany). Between 20 and 30 frustules were used per extraction. The mitochondrial 16S rDNA was amplified by polymerase chain reaction (PCR) using the forward primer (TCGACTGTTTACCAAAAACATA) and reverse primer (ACGGAATGAACTCAAATCATGTAAG) [32]. Each 25 μL reaction included 10–50 ng genomic DNA, 25 pmol of each primer and 62.5 units of OneTaq polymerase (New England Biolabs, Ipswich, MA, USA). Amplifications were performed using the following thermal profile: 5 min at 94 °C; 30 cycles of 5 s at 94 °C; 50 s at 50 °C; 1 min at 72 °C and 10 min at 72 °C. PCR products were screened through gel electrophoresis to verify correct fragment length, and amplicons were cloned using the Invitrogen pCR4-TOPO-TA Cloning Kit (Thermo-Fisher, Waltham, MA, USA) according to the manufacturer’s instructions. Clones were grown overnight in LB medium, and plasmids were purified for sequencing using the QIAprep Spin Miniprep (QIAGEN, Hilden, Germany) following the manufacturer’s instructions. Sanger sequencing using M13F and M13R primers was performed through Azenta (Burlington, MA, USA).

### 2.4. Phylogenetic Analysis

Newly generated 16S sequences from laboratory strains Panama (*C. sowerbii*_PAN); Japan (*C. sowerbii*_JPN); Lawrence, KS, USA (*C. sowerbii*_LKS); Coal City, IL, USA (*C. sowerbii*_CCI) and Inverness, IL, USA (*C. sowerbii*_INI) were sequenced in both directions. Consensus sequences were generated in EMBOSS Cons [33]. These five newly generated sequences along with 29 sequences from GenBank (Appendix A) were aligned with MAFFT using default parameters [34] and manually corrected on MEGA 11 [35]. Phylogenetic analyses were conducted under RaxML [36] using the model GTRGAMMA resulting from Modeltest-NG [37] on the CIPRES portal [38].

### 2.5. Podocyst to Polyp Transition

Cold temperature (4 °C), room temperature (approximately 22 °C) and dry conditions were tested to determine the viability of podocysts and their ability to transform into polyps. The *C. sowerbii*_LKS strain was used for this experiment. Ten podocysts were transferred to a 55 mm petri dish containing HM. Three replicates, for a total of 30 podocysts per treatment, were used. For the cold treatment, podocysts were incubated in HM at 4 °C for 2 weeks. After two weeks, temperature was increased by 3 °C increments approximately every 24 h for one week, for a final temperature of 22 °C. For the room temperature treatment, the podocysts were left at 22 °C for 2 weeks in HM. For the desiccation treatment, podocysts were left to air dry in a petri dish at 22 °C for 2 weeks and then placed into HM. After the stressor period, dishes were observed every 72 h, and the number of podocysts and polyps were counted over the course of three weeks.

### 2.6. Fluorescent Labeling of Nuclei

Replicate experiments were performed as described above, except podocysts were subsampled every week for cell counting. Podocysts were rinsed in phosphate-buffered saline (PBS) and fixed in 4% paraformaldehyde + HM for 15 min, rinsed in PBS + 4% paraformaldehyde for another 15 min at room temperature. Samples were then rinsed in PBS and blocked in PBS-Triton (PBS/0.3% Triton) two times for 5 min each. Podocysts were transferred to a solution of Hoechst 3342 (Thermo-Fisher) (1:200 dilution with PBS-Triton) for 30 min. Samples were rinsed once with PBS-T and mounted with 70% glycerol on slides for fluorescent microscopy. Samples were imaged using the INFINITY ANALYZE 3 microscopy system. Stacked microscopy images were analyzed using Cell Count (ImageJ v.2.14). Images were split into quadrants and the number of cells were counted in one of the quadrants. A total of 3–5 podocysts from each treatment were counted each week for 3 weeks.

### 2.7. Frustule to Polyp and Polyp to Colony Transition

Frustules were obtained from the lab culture strains listed above. Frustules were grown in 55 mm petri dishes or glass bowls containing HM. Effects of temperature on polyp differentiation were examined with four different temperatures (14 °C, 22 °C, 26 °C and 28 °C). Three petri dishes or glass bowls (150 ml) with 10 frustules each for *C. sowerbii*_JPN and *C. sowerbii*_LKS, 15 frustules each for *C. sowerbii*_INI and between 10 and 23 frustules for *C. sowerbii*_PAN were placed in each of the four incubators at the appropriate temperatures. Incubators were kept at a 12 h light/12 h dark cycle. The number of frustules, total polyps and polyps per colony were counted daily over the course of three weeks. For statistical analyses, total polyp counts were adjusted to account for different starting frustules (see below). Frustules produced from polyps during the three weeks were not removed from each replicate dish or bowl; therefore, polyps were formed from the initial frustules as well as from newly produced frustules. Polyps were fed to satiation with *Artemia nauplii* starting at day 5 and continued for 3 times/week. Media was changed after every feeding.

### 2.8. Statistical Analysis and Visualization

Analyses were performed in Rstudio v2024.04.2 + 764 with R 4.4.0. Results were considered significant when *p* < 0.05. The podocyst-to-polyp statistics were performed using a Fisher’s exact test. Bar graphs were generated by ggplot2. Variances in cell count were calculated using standard errors, which were determined by dividing the standard deviation by the square root of the number of samples per week. Specifically, for the podocyst cell count, 3 to 5 podocysts were subsampled each week for staining, which were used to determine the samples for the standard error.

In measuring frustule-to-polyp transitions, the data were first standardized as different strains had different numbers of starting frustules (10 frustules each for *C. sowerbii*_JPN and *C. sowerbii*_LKS, 15 frustules each for *C. sowerbii*_INI and between 10 and 23 frustules each for *C. sowerbii*_PAN). This was accomplished by multiplying the number of polyps by 10 and dividing by the number of starting frustules. Analyses in the transformed data were performed using repeated measures ANOVA. Variances were calculated using standard errors. A Tukey–Kramer post hoc test on the standardized data was used to determine significance. Time series plots were generated by SigmaPlot v.15.0.

## 3. Results

### 3.1. Phylogenetic Diversity of Craspedacusta sowerbii

We reconstructed a phylogenetic tree with 34 16S sequences (5 newly sequenced for this study and 29 from GenBank) (Appendix A) (Figure 3). Clade 1 contains sequences from specimens collected in Europe, North Africa and the newly sequenced clonal strain collected from Inverness, Illinois (*C. sowerbii*_INI). Clade 2 contains sequences from specimens collected in Asia, Europe and N. America, including our newly sequenced clonal strains collected from Cole City, Illinois (*C. sowerbii*_CCI), Panama (*C. sowerbii*_PAN) and Lawrence, KS (*C. sowerbii*_LKS). In addition, we recovered two other distinct lineages, one from specimens collected in China and Japan, including *C. sowerbii*_JPN, and another lineage from an unreported locality (GenBank #MZ569026.1). Our phylogeny was used to determine if strains differed in response to different temperatures and, if so, whether or not this corresponded to their phylogenetic position.

### 3.2. Podocyst to Polyp Transition

Although the exact mechanism(s) for range expansion of *C. sowerbii* into new localities is not known, given that the podocyst stage is the dormant stage, it is the most likely stage responsible for its invasiveness [39]. In order to better understand the conditions in which podocysts can survive and transform into polyps, we subjected podocysts from the *C. sowerbii*_LKS strain to cold temperatures (4 °C) in hydra medium (HM), room temperature (22 °C) in HM and desiccation (room temperature with no media). None of the 30 podocysts that were subjected to dry conditions transformed into polyps, whereas 1/30 in HM at room temperature transformed into polyps and 7/30 in HM subjected to 4 °C transformed into polyps (Figure 4A).

The experiment outlined above was repeated except that podocysts were subsampled and stained with the nuclear stain Hoechst, to count the number of cells within each podocyst. After three weeks, there is a significant decline in cells in the dry treatment (df = 2, F value = 13.17, *p*-value 0.002), whereas in the room temperature treatment, cell number fluctuates (df = 2, F value = 1.82, *p*-value = 0.217 ns), and in cold conditions, the number of cells significantly increases (df = 2, F value = 4.77, *p*-value 0.035) when compared to dry and room temperature conditions (Appendix A), presumably in anticipation of transitioning to a polyp (Figure 4B).

### 3.3. Frustule to Polyp and Polyp to Colony Transitions

In contrast to podocysts, we observe frustules consistently transforming into polyps in our lab cultures at room temperature (approximately 20–22 °C). However, it was unclear whether these were the optimal temperatures for frustule-to-polyp transitions. In addition, given the genetic diversity within the *C. sowerbii* species complex (Figure 3) and the different environments in which they are found, we wanted to determine if different strains differed in response to different temperatures. Specifically, we used laboratory strains that were originally collected from Panama (*C.sowerbii*_PAN), Japan (*C. sowerbii*_JPN), Lawrence Kansas (*C. sowerbii*_LKS) and Inverness Illinois (*C. sowerbii*_INI), which represent three out of the four distinct lineages identified in Figure 3. We tested how many polyps were produced over time, with an initial inoculum of frustules (see Materials and Methods), at 14 °C, 22 °C, 26 °C and 28 °C. During the experimental period, polyps continued to produce frustules, so the increase in polyps were either from the initial frustule inoculation or from frustules budding from newly produced polyps. Using a repeated measure ANOVA, we found significant differences in response to temperature in the *C. sowerbii*_JPN and *C. sowerbii*_LKS strains (Figure 5, Table 1). The maximal average number of frustules that transformed into polyps occurred at 26 °C for all strains (Figure 5) (Appendix A). A post hoc analysis within strains showed a significant difference in polyp production at 26 °C, when compared to the three other temperatures, in the *C. sowerbii*_JPN strain, and when compared to 14 °C and 22 °C in the *C. sowerbii*_LKS strain. The *C. sowerbii*_INI and *C. sowerbii*_PAN strains did not show significant differences (Appendix A). At the conclusion of the experiment, *C. sowerbii*_PAN showed the least variability in polyp production in response to temperature, ranging between 10.2 polyps at 28 °C and 13.1 polyps at 26 °C (Figure 5) (Appendix A). The least number of polyps produced was at 14 °C for all strains (Figure 5, Appendix A).

Although the different strains responded similarly to differences in temperature, the total number of frustules and polyps produced differed between strains. For example, on day 21 at 26 °C, the mean number of polyps produced for *C. sowerbii*_INI was 35.1 (SE 12.4), for *C. sowerbii*_LKS was 20 (SE 2.7), for *C. sowerbii*_JPN was 9.8 (SE 2.7) and for *C. sowerbii*_PAN was 13.5 (SE 1.2) (Figure 5) (Appendix A); although, post hoc analyses did not show significance (not shown).

In addition to the number of polyps produced from frustules, the number of polyps per colony was counted. Primary polyps tend to bud a second polyp more rapidly at higher tested temperatures (26 °C to 28 °C). A post hoc analysis, however, revealed that the only significant difference was in the *C. sowerbii*_LKS strain at 26 °C compared to 14 °C (Appendix A). Towards the end of the 21-day experiment, *C. sowerbii*_INI had a higher average number of polyps at the lower temperature of 14 °C, although without significance (Figure 6) (Appendix A). During the three-week time period, the colony size rarely exceeded two polyps.

### 3.4. Polyp to Medusa Transition

In an effort to experimentally induce medusae budding in the lab, we tested several conditions. These conditions included varying temperature (14 °C, 22 °C, 26 °C and 28 °C), varying media (HM, spring water and DI water) and varying feeding frequency (3X/7days; 1X/14 days) (Table 2). None of these conditions reliably produced increased medusa budding. *C. sowerbii*_LKS never bud medusae in the lab. *C. sowerbii*_PAN and *C. sowerbii*_JPN bud sporadically at 22 °C, 26 °C and 28 °C, but have never been observed to bud medusae at 14 °C. Repeated experiments of starvation for two weeks, followed by feeding, induced medusae budding in *C. sowerbii*_PAN. Medusae only budded in hydra medium and did not bud in distilled water or bottled spring water for *C. sowerbii*_PAN. Interestingly, budding occurred most frequently in the months of July-September in *C. sowerbii*_PAN and *C. sowerbii*_JPN, mimicking seasonal observations in nature [5], even though the temperature in the lab remained relatively constant at approximately 22 °C.

## 4. Discussion

We identified four distinct lineages of *Craspedacusta sowerbii* using 16S data from five newly sequenced clonal strains, combined with available data from GenBank. Our analysis recovered the two distinct lineages reported from 16S sequences in Morpurgo et al. [38] (Clade 1 and Clade 2) (Figure 3). The four lineages recovered are consistent with what was reported in Peterson et al. [30] using the same 16S marker and in Oualid et al. [40] and Morpurgo et al. [41] using different markers. Our experimental strains fell into three of the four identified clades.

Given that podocysts are the dormant stage of *C. sowerbii,* using the *C. sowerbii*_LKS strain, we examined podocyst survival rate and ability to eventually transition into a polyp when exposed to cold conditions, compared to keeping them at room temperature and after desiccation. Podocysts that were desiccated eventually disintegrated after returning to HM. This is contrary to Reisinger [9] who reported that podocysts were resistant to desiccation; although, the author did not provide experimental data to support this assertion. Our results indicate that podocysts from the *C. sowerbii*_LKS strain cannot survive in dry conditions, and thus, the spread of *C. sowerbii* to new environments is likely restricted to aqueous conditions.

Only one out of 30 podocysts transformed into a polyp when kept in HM at room temperature. By contrast, 7 out of 30 podocysts transformed into polyps when they were exposed to 4 °C for 2 weeks and gradually brought back to room temperature (Figure 4), consistent with what has been reported in Dunham [42]. These results suggest that cold temperatures preserve the podocyst’s viability and the transition from colder temperatures to warmer temperatures induces polyp formation. Podocysts are thus well suited to overwintering, and as temperatures rise in the spring, they are likely triggered to develop into polyps.

We show that more polyps are produced from frustules at 26 °C for all four strains (Figure 5), and this does not differ according to their phylogenetic position (Figure 3). This suggests that despite genetic variability in the *C. sowerbii* species complex, all members can survive in a wide range of environments. This observation is consistent with the widespread distribution of collected specimens between and within the *C. sowerbii* lineages (Figure 3).

Marchessaux et al. [27] tested frustule transition to polyps at 19 °C and 29 °C and found that many more polyps were produced at 29 °C, concluding that increased temperature favors polyp formation and is likely due to increased frustule formation at these high temperatures. Although our results are consistent with the findings of Marchessaux et al. [27], our experiments show that the optimal temperature is 26 °C and that the number of polyps produced is less at 28 °C in all strains and significantly so in the *C. sowerbii*_JPN strain (Appendix A). Thus, although warmer waters favor *C. sowerbii* polyp production, there is a limit to this favorability, as the higher temperature of 28 °C is less optimal.

In investigating factors that affect colony size, Marchessaux et al. [27] found that there are more polyps per colony at the lower temperature of 19 °C when compared to 29 °C. Although we did not consistently find lower temperatures favoring a higher number of polyps per colony, our experiments lasted only 21 days as compared to the experiments in Marchessaux et al. [27], which lasted 80 days. Indeed, in our experiments, at day 21 in the *C. sowerbii*_INI strain, the lower temperature of 14 °C had a higher average colony size than the higher temperatures, and the other strains showed a trend towards increasing colony size at lower temperatures towards the end of the experiment, although it did not show significance (Figure 6) (Appendix A). This suggests, however, that had we extended our experiment to 80 days, we might have seen larger colonies at lower temperatures as well.

Unlike frustules, medusae production for *C. sowerbii* is known to be sporadic, budding unpredictably. However, when they are found in nature, they tend to be seasonal, and previous studies have suggested that medusa buds form when temperatures rise [6,23,25,27,43,44]. Although no quantitative data were gathered, we surprisingly also observe seasonal occurrences of medusae budding in the laboratory, even though the temperature remained relatively constant. In particular, we observe an increased frequency of medusa budding in the months of July–September for the *C. sowerbii*_PAN and *C. sowerbii*_JPN strains; although, the *C. sowerbii*_INI strain showed an increased frequency of medusae budding in January-February. Varying conditions, including temperatures, feeding frequency and different growth media, could not reliably induce medusa budding from polyps. Starvation for two weeks followed by feeding appeared to induce budding in *C. sowerbii*_JPN, but not consistently. In particular, newly formed polyps (three-week old) subjected to starvation failed to bud medusae, suggesting that polyps need to mature before acquiring the ability to bud medusae.

Sexual maturation of the medusa stage has never been observed in a lab. We successfully reared medusae for three months where they grew to a diameter of approximately 20 mm and developed immature gonads [3]. Feeding 1-day-old *Artemia* nauplii 5 times/week during the first 2–3 weeks after the medusae were released was more successful at prolonging medusa survival. In addition, we found that after three weeks of growth, transferring the medusae to larger containers with very light aeration, similar to methods reported by Marchessaux and Bejean [28] and Folino-Rorem et al. [3], prolonged growth. Despite these efforts, however, medusae reared in the lab have never spawned gametes. Frequently, as the medusa matured, the bell everted and the medusa died shortly after.

Several studies have looked into prey diversity for *C. sowerbii* medusa [45,46], which could affect *C. sowerbii* maturity during summer and fall months, as different food sources become available. We fed medusae both copepods and *Artemia* nauplii. In contrast to *Artemia*, where we were able to get medusae to survive for three months, we found increased mortality of *C. sowerbii* fed with copepods, starting after 10 days. It appeared that it was more difficult for *C. sowerbii* to catch and ingest the copepods, and thus, the medusae may have died due to starvation. Although copepods are preyed on by *C. sowerbii* [47], the medusa in the lab might have been too small to catch and ingest this prey, or the prey density was insufficient to promote prey capture.

Our study is mostly consistent with previous studies on temperature and food availability and with the seasonal observations for growth and life cycle transitions of *Craspedacusta sowerbii*. Specifically, evidence supports *C. sowerbii* overwintering via podocysts, with cold temperatures preserving viability and warming temperatures triggering transition to the polyp stage. Moderate temperatures (22 °C–26 °C) favor polyp growth, with slightly cooler temperatures favoring colony formation. While we were not able to reliably induce medusae budding at higher temperatures, we found that higher temperatures (≥24 °C) did indeed favor medusae growth. These seasonal variations are likely interconnected with food availability and diet.

It is worth noting that occurrence records of *C. sowerbii* are sparse in tropical regions when compared to higher and lower latitudes. Only 11 out of 3223 occurrence records in the last 60 years are from the tropics [12]. Given the different optimal temperatures for each life cycle stage transition, it is possible that *C. sowerbii* is less successful in regions where there is less variation in temperature (i.e., the tropics) and that it is the seasonal variations in temperature that promotes transitions and growth of each life cycle stage. Further experiments are needed to confirm that seasonal transitions are necessary for life cycle transitions.

## 5. Conclusions

*Craspedacusta sowerbii* is a widespread invasive species whose jellyfish stage appears sporadically. Knowledge of the conditions in which *C. sowerbii* can grow, transition to different life cycle stages and spread is critical for predicting the potential effects of *C. sowerbii* in a warming climate. Here, we show that *C. sowerbii* has at least four genetically distinct lineages based on 16S sequences with our five newly sequenced specimens combined with data available in GenBank. Despite this diversity, lineages across the *C. sowerbii* species complex demonstrate similar responses to growth under different temperatures. Our finding that the dormant podocyst stage is not resistant to desiccation in the *C. sowerbii*_LKS strain, but does require low temperatures to maintain viability, suggests that this life cycle stage is suitable for overwintering, but not suitable to disperse under dry conditions. The sensitivity to temperature changes for life cycle transitions suggests that *C. sowerbii* relies on seasonal changes for completing its life cycle, which in turn may explain its denser distribution and abundance in more northern and southern latitudes as compared to tropical regions where observations have been sparse.

## Figures and Tables

**Figure 1 biology-13-01069-f001:**
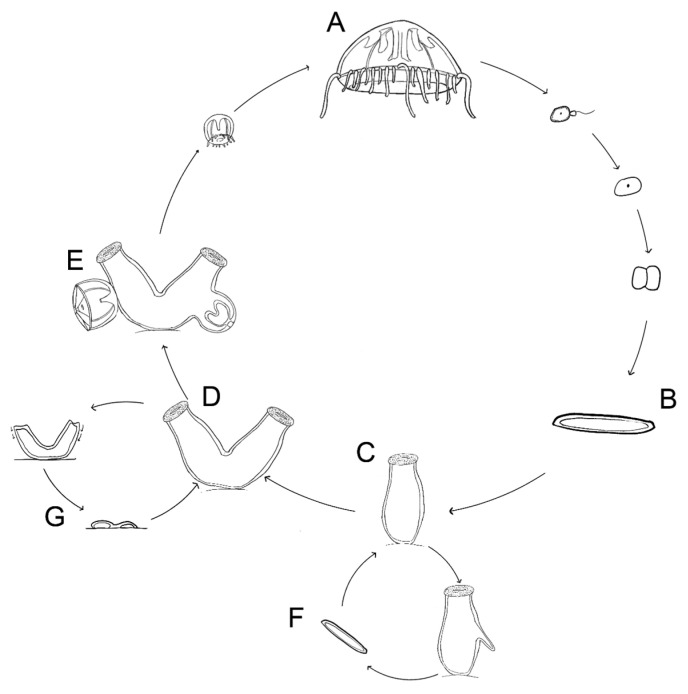
Life cycle of *Craspedacusta sowerbii.* (**A**) Mature medusae; (**B**) planula larva; (**C**) primary polyp; (**D**) two-polyp colony; (**E**) polyp budding medusae; (**F**) frustule; (**G**) podocyst.

**Figure 2 biology-13-01069-f002:**
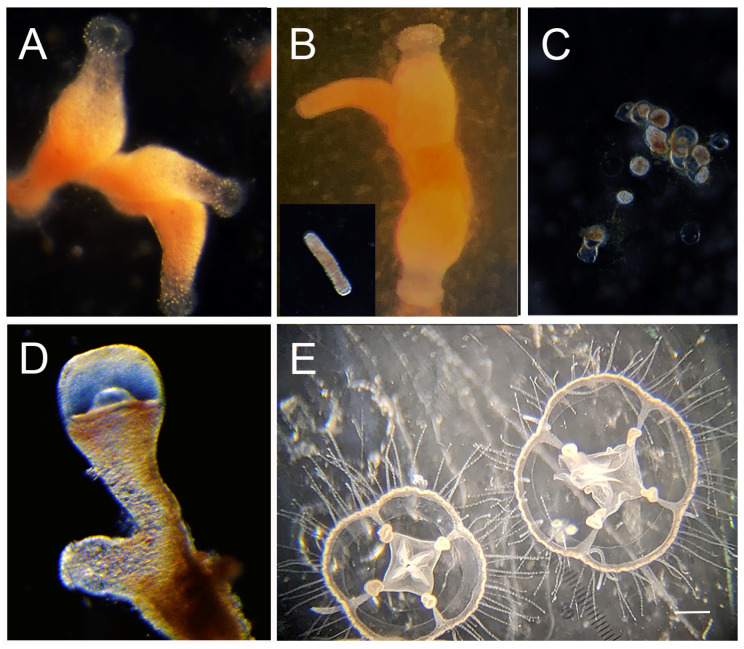
*Craspedacusta sowerbii* life cycle stages. (**A**) A three-polyp colony; (**B**) polyp budding a frustule (inset frustule); (**C**) podocysts; (**D**) polyp budding medusa; (**E**) medusae budded from the *C. sowerbii*_INI strain. m = mouth of polyp, f = frustule budding of polyp and mb = medusa bud. Scale bar for (**E**) = 1 mm.

**Figure 3 biology-13-01069-f003:**
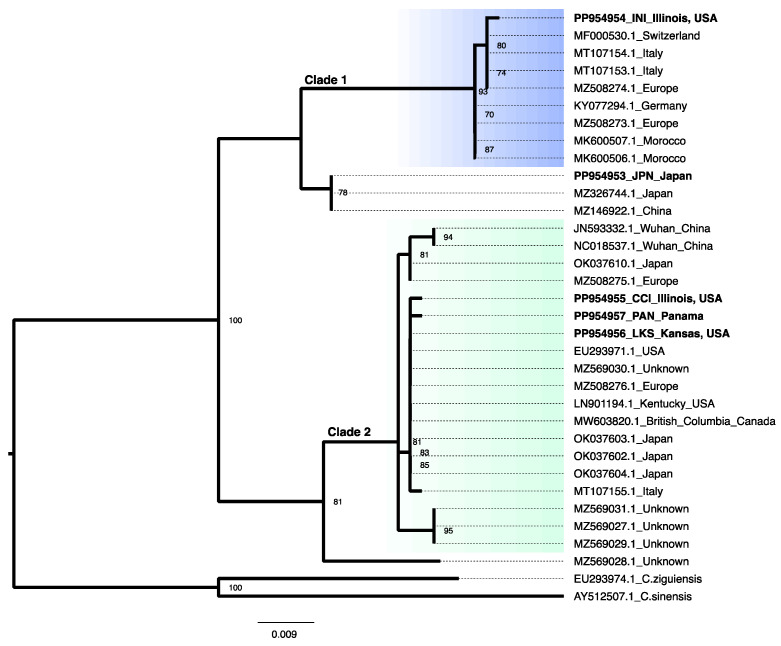
Maximum likelihood phylogenetic analysis using 16S sequences. Taxon labels indicate GenBank Accession numbers and reported localities (Appendix A). Only bootstrap values ≥ 70 are shown. Those in bold are newly generated sequences for this study. Clade 1 and Clade 2 correspond to those clades reported in Morpurgo et al. [29].

**Figure 4 biology-13-01069-f004:**
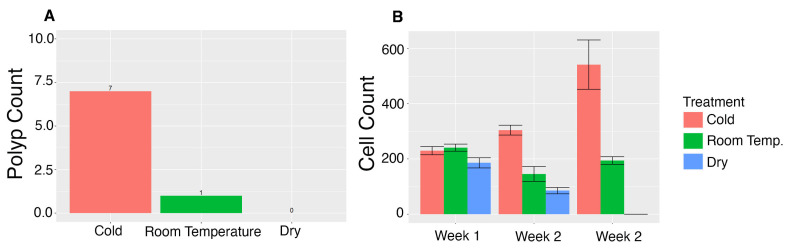
(**A**) Number of polyps that were produced out of 30 podocysts in the different treatments; (**B**) average number of cells in each podocyst at three time points following treatment. The treatments are cold (4 °C), room temperature (22 °C) and dry (desiccation at room temperature).

**Figure 5 biology-13-01069-f005:**
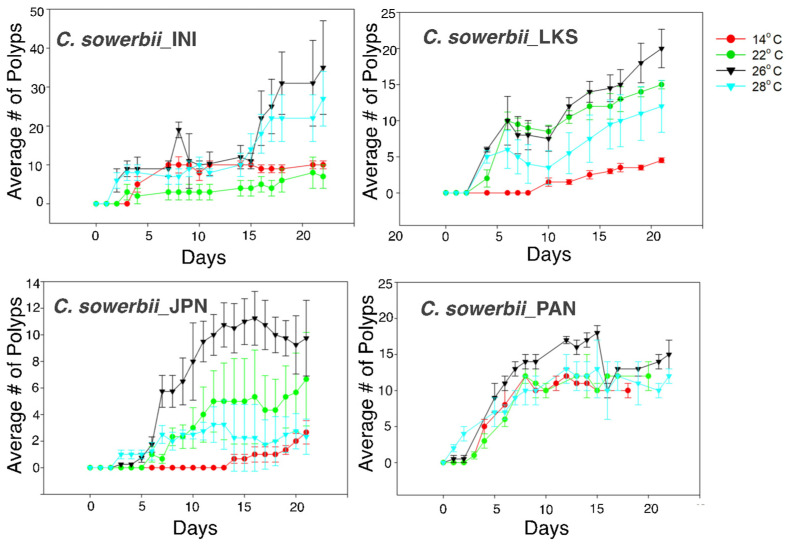
Polyp formation from frustules at different temperatures. Data were standardized to adjust for the different number of starting frustules.

**Figure 6 biology-13-01069-f006:**
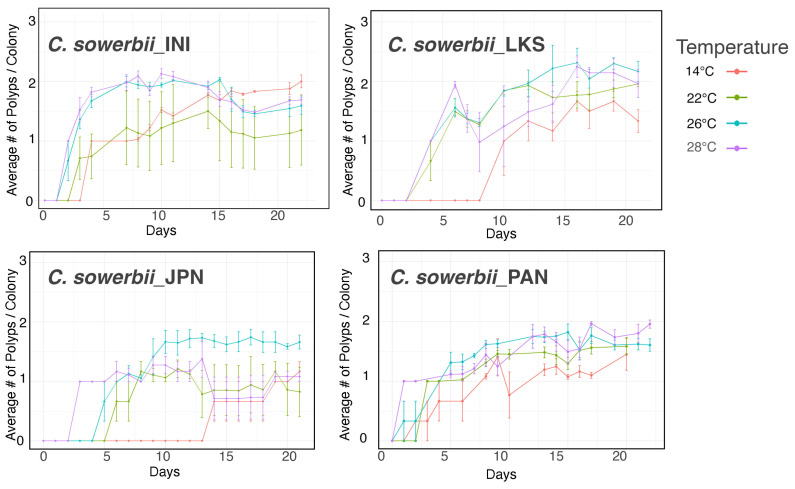
Colony formation from primary polyps at different temperatures.

**Table 1 biology-13-01069-t001:** Results from repeated measure ANOVA in the temperature treatment experiment (see Figure 5). Shown are *p*-values for individual strains.

	F Value	*p*-Value
*C. sowerbii*_INI	47.5	=0.080
*C. sowerbii*_LKS	9.02	<0.001 *
*C. sowerbii*_JPN	7.14	<0.001 *
*C. sowerbii*_PAN	2.2435	=0.161

Degrees of freedom = 3 for all calculations. * Significant values (*p* < 0.05).

**Table 2 biology-13-01069-t002:** Tested conditions and seasonal peaks in which medusae budding has been observed in the laboratory.

Condition	PAN	LKS	INI	JPN
14 °C	−	?	−	−
22 °C	+	?	+	+
26 °C	+	?	+	+
28 °C	+	−	+	+
Increase budding frequency July–September	+	−	−	+
Increase budding frequency January–February	−	−	+	−
14 days of starvation followed by feeding	?	?	?	+
Bottled spring water	?	?	?	−
Distilled water	?	?	?	−
Hydra Medium	?	?	?	+

+ indicates observed medusae budding under stated condition; − indicates that medusae budding has never been observed under stated this condition; ? indicates the strain has not been subjected to stated condition. Unless otherwise specified, all strains were kept at 22 °C in hydra medium and fed 3 times per week.

## Data Availability

Newly generated 16S sequences can be found in GenBank, and accession numbers can be found in the Appendix A.

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
