# Peer review of "Life Cycle Transitions in the Freshwater Jellyfish Craspedacusta sowerbii"

_biology, 2024, doi:10.3390/biology13121069_

Round 1
Reviewer 1 Report
Comments and Suggestions for Authors
Please see the comments of PDF file.
Main concern is the too fewness of data as evidence to argue your discussion.
There are some minor concerning, so I want you to revise your article carefully. The theme itself is interesting, so please polish how to write.

Reviewer 2 Report
Comments and Suggestions for Authors
The MS is both highly relevant and engaging. However, there are several comments and recommendations that should be considered by the authors.
Simple Summary and Abstract
It is not clear what the authors mean by 'optimal growth conditions'. This phrase already appears in the Abstract (15-16) without any explanation. What does 'growth' mean in this case, and what conditions were considered 'optimal' in the study? The sentence 'We found that optimal growth conditions are not correlated with phylogenetic relationships' (15-16) is also unclear. What does this have to do with 'phylogenetic relationships' when the article refers to one species, Craspedacusta sowerbii?
15 - 'different life cycle stages'.
It should be made clear that this does not mean sexual reproduction, only asexual reproduction. It would be good to list these 'different life cycle stages', if not in the simple summary, then in the abstract.
Key words
I recommend that the authors add the words Cnidaria and Hydrozoa.
Introduction
37 - 39 "... including the hydrozoan Cordylophora, the freshwater Hydra, the parasitic Myxozoa and Polypodium and freshwater jellyfish including C. sowerbii and its close relatives."
Not only Cordylophora, but all the organisms listed above (except Myxozoa and Polypodium), belong to the class Hydrozoa.
43 - 44 "small colonies comprising 2-5 polyps"
The colony of Craspedacusta is clearly unusual for hydroids. Judging from the figure, the hydrants (polyps) are connected to each other by the bases of their body columns and no stolons are formed. Is this the case? I suggest that the authors add a few words about the peculiarities of the colony structure.
44 "Polyps bud medusae (jellyfish) from the apical region of the polyp".
According to the drawing, the medusa buds from the middle area of the polyp body column, approximately in the same area where Hydra forms buds. Either the drawing needs to be changed or the description of the budding process needs to be clarified.
58 "Podocysts have been reported to persist under extreme conditions"
The authors need to clarify exactly what are the conditions under which a podocyst can survive? This has direct relevance to the ability of Craspedacusta to spread and is therefore important for this article.
Figure 2
The photos need labelling (body parts of polyps, etc.) as well as scale bars. In the figure 2D it is not clear how many medusae are budding and from which polyp. I recommend replacing this photo with a better quality image. A close-up photo of the podocyst would be good, like the photo of the frustule in 2B.
Materials and Methods
2.5 Podocyst to polyp transition
179 "Three replicates, for a total of 30 podocysts per treatment were used."
How many podocysts were there in each of the three replicates and what were the results for each of the replicates? What is the reason for the low number of podocysts in each replicate?
How were the experimental conditions (especially the temperatures and duration of each treatment) chosen?
2.7. Frustule to polyp and polyp to colony transition
What was the reason for the choice of temperature values for the experiments (14C, 22C, 26C, 28C)?
Results
3.2.1. Polyp to frustules and podocyst transitions
254 - 255 "In the lab, frustules were continually found in our culture dishes, but were especially abundant when polyps were fed at least 3X/week"
The term "especially abundant" is not defined. It would be beneficial to have data on the number of frustules in the control group and under the change in the feeding regime.
256 - 258 "By contrast, podocysts were typically produced under stressful conditions, such as when the polyp culture had been starved for over two weeks, or when algae was abundant, altering the water conditions in the culturing dishes."
The authors need to provide data on the number of podocysts under control conditions and under conditions of stress.
Figure 6. Colony formation from primary polyps at different temperatures.
This figure is not informative, as the number of polyps varies from 1 to 2. In this case it is better to give a table with the average number of polyps per colony.
3.3.5. Polyp to medusa transition
Could you please provide data from the experiment in which medusae formation was obtained? What was the number of polyps in the experimental and control groups? How many replicates were run? How many medusae were formed per 1 polyp in the control and in the experiment? What was the dynamics of medusae formation after the treatment?
3.2.5. Medusa growth
Could you please indicate which gonads were formed in medusae? Were medusae of both sexes formed, only males or only females? Were different strains different in the formation of females and males?
Discussion
369 - 371"Increased feeding of polyps results in polyps of larger size, which likely increases their production of frustules. Decreased feeding of polyps, along with cooling temperatures, likely signals polyps to produce podocysts for overwintering."
It is necessary to provide data in order to give support to these assumptions.
392 - 394 "Indeed we also observe that polyps grow in size faster at increased temperatures, and larger polyps produce more frustules, which in turn make more polyps at a higher rate."
The authors do not present data on polyp growth, nor do they examine the relationship between the number of frustules and the efficiency of their metamorphosis and the size of the polyp.
Reviewer 3 Report
Comments and Suggestions for Authors
The manuscript requires a clear distinction between the "quantitative experiments" presented in the paper and the "qualitative observations”. Furthermore, there are no clear statistical descriptions of the experimental results. I recommend dividing the results chapter into two sections: a) a description of the qualitative observations, and b) the results of the quantitative experiments. Additionally, many observations in the results chapter resemble those typically found in the discussion chapter. It is essential to retain only the descriptions of observations in the results chapter and to relocate the interpretation of the data to the discussion chapter.
Line 156: please insert a reference to the primers (Cunningham and Buss, 1993). The materials and methods section lacks a general schematic representation of the experiments, which would enhance understanding.
Figure 4: please clarify how the variances were calculated.
Lines 280-281: while it is evident that the cell number decreases in the "dry" experiment, this is not as apparent for the "room temperature" experiment. The graph indicates fluctuations in cell number, suggesting variability in this parameter. Please provide the results of the analysis based on the statistical criteria for comparing the data.
Line 299: how was the "optimal temperature" for frustule formation determined? Is there any statistical support for this conclusion? Although Figure 5 indicates that maximum values for the number of frustules occurred at 26 degrees, it remains unclear whether these differences are statistically significant.
Figure 5: it is not evident from the graphs which differences are statistically significant; these should be indicated in a separate table. Additionally, please clarify how deviations from the mean were calculated.
Line 308: how was the "optimal temperature" determined? The graphs display considerable variability in the data and suggest that other factors may influence the experiments. Overall, there are no clear patterns evident in the presented graphs, nor is there a comprehensive statistical analysis of the results.
Figure 6: it remains unclear which differences are statistically significant; this information should also be provided in a separate table. Furthermore, deviations from the mean are not represented in the graph.
Comments on the Quality of English LanguageModerate editing of English language required
Round 2
Reviewer 1 Report
Comments and Suggestions for Authors
OK, the study was improved.
Some point, there are continuation of (), so please use one () and write by ;.
Author Response
Comment: Some point, there are continuation of (), so please use one () and write by ;
Response: I could not find where this exists in teh manuscript. The only example if found is on line 445 ".....(p-value = 0.561) (Figure 5). I do not think it is appropriate to put them within the same brackets as Figure 5 refers to more than just that one p-value.
Reviewer 2 Report
Comments and Suggestions for Authors
The authors extensively revised the manuscript and made changes to improve the clarity of the text and increase confidence in the results of the study. Some of the authors' observations that were not supported by quantitative data and statistical analysis have been moved to the Discussion. The ms is almost ready for publication. However, the Results section still contains statements that are not supported by statistical analysis:
407 - 413 In addition to the number of polyps produced from frustules, the number of polyps per colony were counted. Although we did not extend the experiment long enough for larger colonies to form and did not test for statistical significance, primary polyps appear to bud a second polyp more rapidly at higher tested temperatures (26°C to 28°C). However, towards the end of the 21-day experiment, C. sowerbii_INI primary polyps appeared to bud a second polyp at a higher rate at the lower temperature of 14°C (Figure 6). During the three-week time period, the colony size rarely exceeded two polyps.
Author Response
Comment: 407 - 413 In addition to the number of polyps produced from frustules, the number of polyps per colony were counted. Although we did not extend the experiment long enough for larger colonies to form and did not test for statistical significance, primary polyps appear to bud a second polyp more rapidly at higher tested temperatures (26°C to 28°C). However, towards the end of the 21-day experiment, C. sowerbii_INI primary polyps appeared to bud a second polyp at a higher rate at the lower temperature of 14°C (Figure 6). During the three-week time period, the colony size rarely exceeded two polyps.
Response: We explicitly said we did not test for statistical signficance and instead were showing this because it appeared that there was a trend towards higher budding at lower temperatures as colonies grew larger. We made this more explicit with by modifiying the paragraph pasted below:
In addition to the number of polyps produced from frustules, the number of polyps per colony were counted. Primary polyps appear to bud a second polyp more rapidly at higher tested temperatures (26°C to 28°C). However, towards the end of the 21-day experiment, C. sowerbii_INI primary polyps appeared to bud a second polyp at a higher rate at the lower temperature of 14°C (Figure 6). During the three-week time period, the colony size rarely exceeded two polyps and thus we did not test for statistical significance given that short duration of the experiment and the small number of polyps per colony. This results however shows a potential trend towards higher budding rates at lower temperatures.
Reviewer 3 Report
Comments and Suggestions for Authors
Please expand the chapter “Statistical analysis”.
In lines 278-279, it is stated that “Variances were calculated by standard errors which were generated by dividing standard deviation over the number of samples per week.” It is important to clarify that the standard error of the mean is defined as the standard deviation divided by the <<square root>> of the sample size. Additionally, in the experiment titled “Podocyst to Polyp Transition” it is noted that “Three replicates, for a total of 30 podocysts per treatment were used” (Line 228). Could you confirm whether the average and standard deviation were calculated for these three replicates? Furthermore, it is mentioned that “3-5 podocysts from each treatment were counted each week for 3 weeks” (Line 257). Did you compute the average and variance for the 3-5 podocysts or for the three replicates of the experiment?
In lines 402-406, please specify which pair of experiments (for example, t=22 vs. t=26) corresponds to the reported p-values. Which statistical test was employed to obtain these p-values: repeated measures ANOVA or Fisher's exact test?
Author Response
Comment:
Please expand the chapter “Statistical analysis”.
In lines 278-279, it is stated that “Variances were calculated by standard errors which were generated by dividing standard deviation over the number of samples per week.” It is important to clarify that the standard error of the mean is defined as the standard deviation divided by the <<square root>> of the sample size. Additionally, in the experiment titled “Podocyst to Polyp Transition” it is noted that “Three replicates, for a total of 30 podocysts per treatment were used” (Line 228). Could you confirm whether the average and standard deviation were calculated for these three replicates? Furthermore, it is mentioned that “3-5 podocysts from each treatment were counted each week for 3 weeks” (Line 257). Did you compute the average and variance for the 3-5 podocysts or for the three replicates of the experiment?
In lines 402-406, please specify which pair of experiments (for example, t=22 vs. t=26) corresponds to the reported p-values. Which statistical test was employed to obtain these p-values: repeated measures ANOVA or Fisher's exact test?
Response: Thank you for this comment. We revised the paragraph to clarify and it is pasted below:
Analyses were performed in Rstudio v2024.04.2+764 with R 4.4.0. Results were considered significant when p < 0.05. In measuring frustule to polyp transitions and podocyst to polyp cell counts, statistics were performed using repeated measures ANOVA. The podocyst to polyp statistics was performed using a Fischer’s exact test. Time series plots were generated by SigmaPlot 15.0 and bar graphs were generated by ggplot2. Variances were calculated using standard errors, which were determined by dividing the standard deviation by the square root of the number of samples per week. Specifically, for the podocyst cell count, 3 to 5 podocysts were subsampled each week for staining, which were used to determine the samples for the standard error.
Round 3
Reviewer 2 Report
Comments and Suggestions for Authors
The authors have made the necessary corrections. The revised version of the paper is ready for publication.
Author Response
The reviewer did not suggest any further revisions. We thank this reviewer for their time in improving this manuscript.
Reviewer 3 Report
Comments and Suggestions for Authors
Accept in present form
Author Response

(The authors gave the same response as above.)
